# Computational Model for Early-Stage Aortic Valve Calcification Shows Hemodynamic Biomarkers

**DOI:** 10.3390/bioengineering11100955

**Published:** 2024-09-24

**Authors:** Asad Mirza, Chia-Pei Denise Hsu, Andres Rodriguez, Paulina Alvarez, Lihua Lou, Matty Sey, Arvind Agarwal, Sharan Ramaswamy, Joshua Hutcheson

**Affiliations:** 1Department of Biomedical Engineering, Florida International University, Miami, FL 33174, USA; chsu@fiu.edu (C.-P.D.H.); arodr1162@fiu.edu (A.R.); palva060@fiu.edu (P.A.); sramaswa@fiu.edu (S.R.); 2Department of Mechanical Engineering, Florida International University, Miami, FL 33174, USA; llou@fiu.edu (L.L.); msey002@fiu.edu (M.S.); agarwala@fiu.edu (A.A.)

**Keywords:** aortic valve, fluid–structure interaction, nanoindentation, hydrodynamics, hemodynamics, tissue engineering, CAVD, PSIS, bioscaffold

## Abstract

Heart disease is a leading cause of mortality, with calcific aortic valve disease (CAVD) being the most prevalent subset. Being able to predict this disease in its early stages is important for monitoring patients before they need aortic valve replacement surgery. Thus, this study explored hydrodynamic, mechanical, and hemodynamic differences in healthy and very mildly calcified porcine small intestinal submucosa (PSIS) bioscaffold valves to determine any notable parameters between groups that could, possibly, be used for disease tracking purposes. Three valve groups were tested: raw PSIS as a control and two calcified groups that were seeded with human valvular interstitial and endothelial cells (VICs/VECs) and cultivated in calcifying media. These two calcified groups were cultured in either static or bioreactor-induced oscillatory flow conditions. Hydrodynamic assessments showed metrics were below thresholds associated for even mild calcification. Young’s modulus, however, was significantly higher in calcified valves when compared to raw PSIS, indicating the morphological changes to the tissue structure. Fluid–structure interaction (FSI) simulations agreed well with hydrodynamic results and, most notably, showed a significant increase in time-averaged wall shear stress (TAWSS) between raw and calcified groups. We conclude that tracking hemodynamics may be a viable biomarker for early-stage CAVD tracking.

## 1. Introduction

The human heart beats approximately 70 times per minute, totaling 37 million times a year, and almost 2.8 billion times in an average person’s life. This highly dynamic environment surrounding the aortic valve makes it susceptible to remodeling due to frequent changes in fluid and mechanical stresses. One of the most adverse modes of this remodeling is the manifestation of calcific aortic valve disease (CAVD), which is the most prevalent valve disease globally, affecting an estimated 12.6 million people as of 2017 [1]. CAVD is characterized as a chronic progressive disease in which sclerosis of valve tissue leads to leaflet calcification and ultimately results in stenosis of the aortic valve. Currently, there are no treatment options, pharmaceutical or mechanical, that restore the full functionality of the valve’s leaflets. In fact, the only known treatment for CAVD involves prosthetic valve replacement through either surgical aortic valve replacement (SAVR) or transcatheter aortic valve replacement (TAVR). Aortic valve sclerosis, an early form of CAVD, manifests as mild thickening or stiffening of the aortic valve but generally does not present any large changes to hydrodynamic metrics, such as pressure drop (ΔP), effective orifice area (*EOA*), root mean square flow rate ($Q_{RMS}$), or regurgitant fraction (*RF*), as assessed by conventional transthoracic echocardiography. This means that early-stage CAVD cannot currently be easily diagnosed in the clinic. For example, classification for even mild CAVD is a Δ*P* greater than 20 mmHg, whereas sclerosis typically manifests as a relatively benign Δ*P* of a few mmHg, which is indistinguishable from one for an otherwise healthy valve [2]. As such, CAVD is usually only identified after the disease has progressed to the point where it leads to adverse symptoms, such as shortness of breath, fatigue, or chest pain. At this stage, prosthetic valve replacement is the only available recourse.

However, hemodynamically, it has been found that abnormal wall shear stresses (*WSS*), which are often reported as time-averaged WSS (*TAWSS*) across a cardiac cycle, promote pro-inflammatory pathways that lead to an early, abnormal valve tissue remodeling response, which is a precursor to CAVD [3,4]. Similarly, oscillations in blood flow, which are measured using another hemodynamic parameter known as the oscillatory shear index (*OSI*) [5], first defined by He and Ku, is another parameter in valve hemodynamics. *OSI* has been shown to be involved in leaflet cell alignment [6], with specific *OSI* values promoting the valvular phenotype [7]. Under high *OSI* conditions, it results in increased interstitial aortic valve cell calcification [8]. Compared to normal healthy aortic valve tissue [9], severely calcified aortic valves are often mechanically stiffer due to the presence of microcalcifications [10], which are morphologically similar to atherosclerotic plaques in arteries [11].

A growing trend in cardiovascular treatment development is the use of bioscaffolds to mimic the structural behavior of native cardiovascular tissues. In cardiovascular applications, porcine small intestinal submucosa (PSIS) has been used as a bioscaffold used for cardiac patches [12], tricuspid valve [13], and mitral valve procedures [14]. With PSIS serving as a heart valve bioscaffold, it can also be seeded with interstitial and endothelial cells to mimic native aortic valve cell populations.

Thus, in this study, we used a PSIS scaffold as a testing environment to investigate whether there are any hydrodynamic or mechanical differences between an uncalcified and very mildly calcified valve. In addition, we conducted fluid–structure interaction (FSI) simulations on computer-aided design (CAD) reproductions of these valves to ascertain any measurable hemodynamic changes, such as TAWSS and OSI, between groups. Despite clinical data suggesting there should be no significant differences in any hydrodynamic metrics between a normal aortic heart valve and one with early levels of calcifications, we suspect there may be measurable differences in hemodynamics instead, which could lead to the identification of a possible fluid-based biomarker for early-stage CAVD, as has been found in arteries for atherosclerosis.

## 2. Materials and Methods

### 2.1. PSIS and Valve Sample Preparation

PSIS bioscaffolds were obtained from CorMatrix (Roswell, GA, USA) and used for all valve testing. Three groups (n = 3/group) were used in this study: raw PSIS (no cells seeded onto bioscaffold), PSIS seeded with Human Valvular Interstitial Cells (VICs) and Human Valvular Endothelial Cells (VECs) in a static environment, and PSIS seeded with Human VICs/VECs under dynamic oscillatory flow culture. Human VICs and VECs plus respective culture media were acquired from Innoprot (Innoprot, Derio, Spain) and Lonza (Lonza, Basel, Switzerland), respectively, and were then expanded until passage 4 where the cells were at a proper seeding density of 2 million cells per 2.5 cm^2^ and a VIC to VEC ratio of 27:34 to best mimic the native extracellular matrix (ECM) of the aortic valve in terms of strength and elasticity [15]. To minimize contamination, the cell culture environment was sterilized with 70% ethanol. The bioscaffolds were gas sterilized with ethylene oxide for 12 h prior to the start of valve seeding. The cultured cells were seeded onto the PSIS valve for a total of 7 days, with the human VICs being seeded first. After the 4th day, the VECs were then added, and the co-culture continued until day 7. The PSIS valves were then transferred into either a static or a bioreactor-conditioned environment and conditioned with pro-calcific media. The pro-calcific media consisted of DMEM, 5% FBS, 1% P/S, 1.8 mM of CaCl_2_, 3.8 mM of NaH_2_PO_4_, and 0.4 units/mL of inorganic pyrophosphate [16]. The static condition indicated a no-flow environment while the bioreactor condition was operated at the maximum oscillatory flow condition of 50% positive and 50% negative flow, with an OSI of 0.5 and a shear stress of 3 to 4 dynes/cm^2^, which has been used in previous valve studies [17]. Valves in both the static and bioreactor groups remained in their respective pro-calcific environments for a total of 7 days before being removed for subsequent testing. Cellular activities were recorded in alizarin red staining (ARS) images as well as immunofluorescence (IF) stains. The red dye from ARS suggests the presence of calcification due to cellular activity (Figure 1D–F), and the upregulation of αSMA from IF images also indicates cellular activity on the bioscaffold (Figure 1G,H).

### 2.2. Hydrodynamic Testing

PSIS valve functionality was tested using the ViVitro Pulse Duplicator system (ViVitro Labs, Inc., Victoria, BC, Canada), which has the capability for recording flow and pressure data around any valve being tested. It has been widely used in the field of mechanical, native, and bioprosthetic heart valve testing [18]. The prepared valve samples were sutured onto 26 mm holders which had been 3D-printed (Flashforge Finder, Zhejiang Flashforge 3D technology Co, Jinhua, Zhejiang, China) with polylactic acid (PLA) filament (Figure 1A–C). Sutures along the three posts induced a tri-leaflet shape before being placed in the aortic valve position in the pulse duplicator along with a bi-leaflet mechanical valve in the mitral position. The three valve groups tested were (1) raw PSIS, (2) statically cultured PSIS bioscaffold with seeded VICs/VECs, conditioned in pro-calcific media, and (3) bioreactor high-oscillatory-flow-conditioned (OSI = 0.5) and cultured PSIS bioscaffold with seeded VICs/VECs, in pro-calcific media, with all groups having 3 valves each, (*n* = 3). All valves tested were at 20 °C in a saline solution, 0.90% *w*/*v* NaCl, with a mean arterial pressure of 100 mmHg, a heart rate of 70 beats per minute, and a flow waveform that was 35% systolic. Ventricle and aortic pressure as well as transaortic flow were recorded and averaged across 10 waveform cycles for later analysis. Video recording of the valves in the aortic position was also performed using a Chronos 1.4 High-Speed Camera (Kron Technologies Inc., Burnaby, BC, Canada). Post-processing was performed in MATLAB 2022a (MathWorks, Inc., Natick, MA, USA) using a custom in-house script. Through this functional hydrodynamic testing, the following four metrics were calculated for evaluation: RF, ΔP, Q_RMS_, and EOA.

### 2.3. Calcium Histological Assay

Following hydrodynamic functionality testing, all valve groups were sliced into strips and samples were fixed in 10% formalin at 4 °C for 24 h and then submerged in OCT gel (Scigen Tissue-Plus™ O.C.T. Compound, Thermo Fisher Scientific, Waltham, MA, USA) and frozen at −80 °C for later sectioning. Using a cryostat (Leica CM3050 S, Leica Biosystems, Deer Park, IL, USA), the samples were cut at 16 μm depths and prepared for histological viewing. To confirm tissue calcification, alizarin red staining (RICCA Chemical Company, Arlington, TX, USA) was performed. Alizarin red is a chemical compound that binds to calcium and is often used to qualitatively assess calcification in cultured tissues [8,19,20]. Histological data were then analyzed qualitatively to determine both the presence and degree of calcification between the raw, static, and bioreactor groups.

### 2.4. Nanoindentation

Mechanical testing was performed at 20 °C using a nanoindenter (BioSoft™ In Situ Indenter Bruker, Billerica, MA, USA) with a 500 μm diamond flat-end tip (Bruker, Billerica, MA, USA). Each PSIS tissue strip group, raw, static, and bioreactor, was tested twice at the middle of the sample with 3 samples per group (*n* = 3). Each sample was cut into rectangular sections of 10 × 10 mm, with their thicknesses being 0.4, 0.7, and 0.5 mm for raw, static, and bioreactor groups, respectively. A quantity of 2 mL of PBS was added to the sample surface before indentation. The experiment began once the probe achieved tissue contact and consisted of a total of 3 segments: a loading section of 40 μm displacement at 2 μm/s, an unloading period of −40 μm displacement at −2 μm/s, and lastly, retraction at −5 μm/s for 5 s. Data were then processed through an in-house MATLAB script that fit the measured force and strain during the loading cycle to retrieve Young’s modulus, or stiffness, values using a Hertzian contact model (1):(1)P=4ERδ323(1−ν2)
where P is the applied load from the tissue, R is the indenter diameter, δ is the indentation depth, E is the elastic or Young’s modulus, and v is the Poisson ratio, which was set to 0.30.

### 2.5. Geometric Modeling and Meshing

SolidWorks 2020 (Dassault Systèmes, Vélizy-Villacoublay, France) was used to reconstruct only the aortic valve conduit of the ViVitro Pulse Duplicator to reduce computational complexity. In a similar vein, suture holes of the 3D-printed 26 mm valve holder as well as the threads used were omitted. Inlet and outlet sections were extended to 2× and 3× valve diameter, respectively, to mitigate boundary-induced flow effects (Figure 2A). Prior to meshing, the surface geometry was exported to ANSYS SpaceClaim 2022 R2 (ANSYS, Inc., Canonsburg, PA, USA) for model cleanup, such as the removal of sliver faces, short edges, patching holes, and axis reorientation. Solid and fluid surfaces were meshed in ANSYS LS-PrePost V4.11 with fully integrated elements with an average surface mesh skewness of 0.20 and orthogonal quality of 0.82. For the fluid volume mesh, three boundary layers were expanded on the wall to better capture the wall flow metrics and Delaunay criteria were used to initialize the volume mesh with space-filling tetrahedral elements (Figure 2C). A mesh independence study was performed with a maximum element size ranging from 0.25 to 2.00 mm for both solid and fluid domains. After an analysis of a quasi-static one-way FSI simulation at peak systolic flow, 1 mm element size was chosen as it showed max velocity and maximum von Mises fibrosa side stress having <5% change compared to the finest element size of 0.25 mm (Table 1). Large mesh deformations in aortic valve FSI simulations performed using ALE formulations are expected, which can result in highly skewed elements and, if not treated, convergence issues. To mitigate this, automatic mesh refinement was performed once mesh orthogonal quality fell below 0.8 or an element inverted anywhere in the domain.

### 2.6. Fluid Domain

The pressure differential between the ventricle and aortic chambers, taken from pressure transducers during in vitro testing (Figure 2B), was prescribed on the inlet with 0 mmHg imposed on the outlet. All other surfaces were treated as no-slip walls. The fluid, saline solution, is an incompressible, Newtonian solution with an assumed constant density of 1007 kg/m^3^ and dynamic viscosity of 1.07 cP. Based on initial flow metrics calculations, Reynolds numbers were found to be in the transitory region, ~9800; therefore, turbulence was modeled using a variational multiscale approach, which uses the subgrid scale approach to decompose the fluid domain into regions where small length scales are modeled while larger ones are captured by the mesh [21]. Automatic time-stepping was set with a max time-step of 3 × 10^−4^ s to satisfy the Courant–Friedrichs–Lewy (CFL) condition of 0.5. Residuals for momentum, pressure, and turbulence equations were set to 10^−5^. Three cardiac cycles of 0.854 s each were simulated to ensure cyclic independence and the results of the third cycle were used for analysis.

### 2.7. Solid Domain

While the acrylic aortic valve conduit and a 3D-printed valve holder were included in the model, they were constrained in all motion and rotation to simplify the analysis. As such, the only material undergoing deformation was the bioscaffolds, which were all modeled as isotropic, homogeneous, linear elastic materials. Young’s moduli for the elastic property were taken from the nanoindentation analysis. Each sample had two nanoindentation recordings, so their sample averaged values were used during analysis. A Poisson’s ratio of 0.3 was set for all groups. Similarly, leaflet thicknesses were set to 0.4, 0.7, and 0.5 mm for raw, static, and bioreactor groups, respectively. While sutures and their respective holes were omitted from the geometry, their effects were included as fixed nodes on the scaffold surface to induce a tri-leaflet shape during coaptation. The inlet and outlets of the conduit were also fixed in XYZ and in all rotations.

### 2.8. Fluid–Structure Interaction

Strong two-way FSI coupling was performed using ANSYS LS-DYNA’s ICFD and implicit mechanical solvers whereby pressure and displacement information is passed between them. To promote solution stability the fluid time-step was enforced between solvers as it was the smaller of the two. The solver uses the arbitrary Lagrangian–Eulerian (ALE) method to solve the flow and structural domains. With the Lagrangian approach, nodal values are tracked through a domain, making it preferred for structural analysis. Eulerian methods use a fixed grid where the flux through each element is tracked, making it popular for fluid analysis. In the ALE approach, nodes are allowed to move in a Lagrangian fashion following the local velocity; this allows for velocity and stress to propagate through time. However, since this distorts the mesh, a relaxation step is applied with a Lagrangian smoothness operation. The results from the previous mesh are then remapped to the new mesh and the analysis continues. Under the ALE configuration the classic Navier–Stokes equations for momentum and continuity governing fluid flow are:(2)$$\rho_{f}\left(\frac{\partial \vec{u_{f}}}{\partial t}+\left(\vec{u_{f}}-m_{f}\right)\cdot \nabla \vec{u_{f}}\right)=-\nabla P+\mu \nabla^{2}\vec{u_{f}}+\rho_{f}\vec{g}$$
(3)$$\nabla \cdot u_{f}=0$$
where $\rho_{f}$ is the fluid density, $\vec{u_{f}}$ is the fluid velocity vector, $m_{f}$ is the velocity, ∇ is the divergence operator, P is the fluid pressure, μ is the dynamic viscosity, and $\vec{g}$ is the body acceleration per cell volume due to gravity. The solid domain is governed by the momentum equations of motion:(4)$$\nabla \cdot \sigma_{s}+{\rho_{s}b}_{s}=\rho_{s}\frac{d\vec{u_{s}}}{dt}$$
where $\sigma_{s}$ is the solid stress tensor, $\rho_{s}$ is the solid density, $b_{s}$ is the body force per cell volume, and $\frac{d\vec{u_{s}}}{dt}$ refers to the local acceleration of the solid domain. The previously described fluid and solid domain are coupled at the scaffold interface, $\Gamma_{f/s}$, using a set of conditions; they are as follows: (1) fluid and solid displacements must be identical, (2) traction vectors at the boundaries must match, and (3) the no-slip fluid boundary condition must remain true:(5)$$\vec{u_{f}}=\vec{u_{s}}$$
(6)$$\vec{\sigma_{f}}{\hat{n}}_{f}=\vec{\sigma_{s}}{\hat{n}}_{s}$$
(7)$$\vec{u_{f}}{\hat{n}}_{f}=0$$
where $\vec{\sigma_{f}}$ and $\vec{\sigma_{s}}$ are the fluid and solid stress tensors. Contact between the leaflets was assumed to be frictionless and was enforced using a penalty-based approach provided by the *CONTACT_AUTOMATIC_SINGLE_SURFACE_MORTAR card whereby numerical springs act to prevent node penetration from one surface with another as well as allowing for transfer of forces between parts. High-frequency oscillations of the leaflets during diastole were reduced using mass-weighted damping, also known as Rayleigh damping, with an α=0.1 m/s [22]. Each group’s specific inlet pressure differential boundary condition, scaffold thickness, and Young’s modulus were prescribed. All other factors, such as the model base geometries, fluid and solid meshes, and simulation parameters, were the same for all groups.

### 2.9. Hemodynamic Parameter Calculations

*TAWSS* was defined to measure the fluid-induced friction on a surface across a cardiac cycle.
(8)$$TAWSS=\frac{1}{T}\int_{0}^{T}\left|\vec{\tau_{w}}\right|dt$$
Here, T is the period of the cardiac cycle and $\tau_{w}$ is the instantaneous wall shear stress on a fluid surface, which can further be defined as:(9)$$WSS=\vec{\tau_{w}}=\mu \frac{\partial \vec{u_{f}}}{\partial {\hat{n}}_{f}}$$
where ${\hat{n}}_{f}$ is the fluid unit normal vector of the surface. Similarly, *OSI* was defined to characterize the degree of flow reversal on a surface due to pulsatile flow.
(10)$$OSI=\frac{1}{2}\left(1-\frac{\left|\int_{0}^{T}\vec{\tau_{w}}dt\right|}{TAWSS}\right)$$
Its value can range from 0 to 0.5, where 0 indicates no exact shear-induced flow direction, while 0.5 is purely oscillatory flow.

### 2.10. Geometric Orifice Area Calculation

The Geometric Orifice Area (GOA) is defined as the cross-sectional open area created between the valve annulus and its leaflets during peak systole, when the valve is fully open. It is often found using planimetry of any number of imaging modalities, such as transthoracic echocardiography, transesophageal echocardiography, cardiac magnetic resonance imaging, or computed tomography (CT) [23,24,25,26]. Modern methods, however, increasingly are relying on ultra-high-resolution CT devices for more accurate cardiac calcium scoring and valve area identification [27,28,29]. Video recordings obtained from hydrodynamic testing were exported and matched with concurrently collected flow data to identify frames associated with peak flow. These frames were then led through a series of image processing steps, using MATLAB’s Image Processing Toolbox, before identification of the open valve area, as follows: grayscale conversion, contrast limited adaptive histogram equalization, Gaussian blurring, graph cutting of the region of interest [30], and then active edge-based contouring [31].

### 2.11. Statistics

All data collected were tested for normality with the Skewness-Kurtosis test (*p* < 0.05). Unpaired Student’s *t*-tests were then performed between the three groups, raw, static, and bioreactor, for each of the data metrics recorded. Specifically, for the hydrodynamic results, the parameters evaluated were *EOA*, *RF*, ∆P, and $Q_{RMS}$. For nanoindentation testing, Young’s modulus values were evaluated. For hemodynamics, *TAWSS* and *OSI* found from FSI simulations were evaluated. In all cases, results were deemed significant if the *p*-value between group means was found to be <0.05, with further levels of significance set at 0.01, 0.001, and 0.0001.

### 2.12. Computational Information

Simulations were conducted on an HP Workstation with a 4-core CPU at 2661 MHz, 2.67 GHz Intel^®^ Xeon^®^ X5550 (Intel Corporation, Santa Clara, CA, USA) with an NVIDIA Quadro FX 1800 (Nvidia Corporation, Santa Clara, CA, USA) with 16 GB of RAM. FSI post-processing was performed through MATLAB 2022a scripting and ANSYS EnSight 2022 R2.

## 3. Results

### 3.1. Calcium Assay: Qualitative Differences between Valves

Histological images from alizarin red staining are shown in Figure 1; visually, there was not a large difference between the calcified static valve (Figure 1E) compared to the calcified oscillatory group (Figure 1F). Both groups exhibited similar levels of calcification based upon the intensity of the staining. However, as expected, the raw PSIS valve group (Figure 1D) did not show any signs of calcification, as indicated by the lack of red staining.

### 3.2. Hydrodynamic Differences and Similarities

Hydrodynamic data were analyzed and used to create a statistical comparative bar chart of functionality parameters relating to cardiac function, namely, *RF*, Δ*P*, *Q_RMS_*, and *EOA*, between each of the three valve groups (Figure 3A–D, respectively). There was no statistical difference between any of the groups for *RF*, *EOA*, or *Q_RMS_*. Δ*P* only showed a statistically significant difference between static and bioreactor groups at a *p*-value of <0.01.

### 3.3. Mechanical Differences in Stiffness

The mechanical results from nanoindentation testing can be seen in Figure 4A–C, showing displacement against load curves for raw, static, and bioreactor groups, respectively, along with an inset showing the indenter contacting the tissue sample. Statistical summary results for Young’s modulus values are shown in a comparative bar graph (Figure 4D). Averaged results were 55.10, 84.79, and 61.72 kPa for raw, static, and bioreactor groups, respectively. All groups showed a statistically significant difference in stiffness values, *p*-value < 0.01, compared against each other.

### 3.4. Hemodynamic Differences and Computational Hydrodynamic Similarities

Velocity and instantaneous *WSS* slice profiles across the cardiac cycle are shown in Figure 5A. Velocity during peak systole for raw static and bioreactor groups was 1.35, 1.68, and 1.45 m/s, respectively. Similarly, *TAWSS* was found to be highest in the bioreactor group, 15.90 dynes/cm^2^, compared to raw, 10.74 dynes/cm^2^, with static being comparable to bioreactor, 15.20 dynes/cm^2^. On average, *WSS* was lowest in the belly region and higher towards the free edge or sinus wall. OSI varied only slightly between groups of raw, static, and bioreactor, with values being 0.46, 0.43, and 0.44, respectively. While there was a statistically significant difference between *TAWSS* of raw with either calcified groups there was not one between static and bioreactor groups (Figure 6A). Despite the slight reduction in *OSI*, there was no significant difference found between any groups for this *OSI*, despite the slight reduction that was seen between raw and calcified groups (Figure 6B). Vorticity analysis using the Q-criterion showed greatest disturbed flow during peak flow for both calcified groups with only smaller vortices found at the leaflet free edge in the raw case (Figure 5B). As the FSI solves for both fluid pressure and velocity, the previously attained in vitro results were compared with the FSI in silico model for validation purposes (Table 2); values shown are averages across three samples per group. Comparisons across hydrodynamic outcomes of the FSI model for Δ*P*, *Q_RMS_*, and *EOA* were less than 30% different from the in vitro results. However, *RF* was found to be substantially higher across all groups.

### 3.5. Valve Area during Peak Flow

Valve areas were calculated for the peak systolic time-point corresponding to each valve group. Both in vitro and in silico peak systolic frames are shown in Figure 7 alongside a summarized table of GOA values shown in Table 3. Raw PSIS had the greatest GOA in both in vitro imaging and in silico FSI simulations of 3.24 and 3.08 cm^2^, respectively. Calcification of the valve reduced the in vitro and in silico GOA, with the static case GOA being 2.26 cm^2^ and 2.76 cm^2^, respectively. Similarly, the bioreactor group showed a slightly higher GOA of 2.83 cm^2^ and 2.96 cm^2^, respectively. In all groups, the in vitro results were always greater than in silico; however, there was no greater than a 20% difference between the in vitro and in silico values. The valve mobility and GOA were greater in the in vitro recordings than in our simulations; this is likely due to the use of high-frequency dampers added for model stability.

### 3.6. Literature Validation

In addition to the in vitro validation of hydrodynamics for our FSI model, we also compared our results with the available literature. The work of Ramaraj and Sorrell had included pulsed wave Doppler echocardiograms of patients with healthy valves showing peak velocities and pressure gradients of 1 m/s and 4 mmHg, respectively [32]. Rezaeian et al. conducted an investigative study of patients scheduled for aortic valve replacement; one aspect of their study used transthoracic echocardiography of 20 control patients to calculate an average healthy *EOA* of 0.95 cm^2^ [33]. A similar 2D aortic valve computational study by Kivi et al. found pressure gradients for healthy valves corresponding to ~7.5 mmHg when validated with their in-house experimental pulse duplicator [34]. Little could be done for additional validation of our early-stage calcified valve, or aortic valve sclerosis, as valves at this stage are known to be asymptomatic and, therefore, their hydrodynamics are comparable to a naturally healthy valve even though they can progress to aortic stenosis [35]. Overall, our results summarized in Table 2 agree well with the literature results of a healthy aortic valve.

Regarding hemodynamics, only studies with healthy valves and those with stages of CAVD (mild, medium, and severe) were found in the literature. A study on calcification grading’s effects under helical inflow by Daryani et al. found that a healthy aortic valve’s fibrosa has a range of *TAWSS* between 0 and 30 dynes/cm^2^ and high *OSI*, nearing 0.5, across the entire surface [36]. They found that the belly had the lowest TAWSS and highest *OSI*, while near the free edge, *TAWSS* was highest and OSI the lowest. Similarly, for a mildly calcified valve, they found *TAWSS* ranging from 0 to 40 dynes/cm^2^ and *OSI* being around 0.5 across the surface, excluding lower values near the free edge [36]. No studies could be found to compare with our very mildly calcified valves as they are hydrodynamically seen as indistinguishable from a healthy valve. Overall, our results had *TAWSS* and *OSI* values that agreed well with the available literature for a healthy valve and showed calcified results lower than those for the mildly calcified valve classification.

## 4. Discussion

The present study conducted hydrodynamic, mechanical, histological, and computational assessments on three valve groups: raw PSIS, statically cultured PSIS bioscaffold valves with seeded VICs/VECs, in pro-calcific media, and bioreactor-based high-oscillatory-flow-conditioned (OSI = 0.5) cultured PSIS bioscaffold valves with seeded VICs/VECs, in pro-calcific media. This let us investigate the degree to which the pro-calcific media affected each valve group’s hydrodynamic functionality, mechanical properties, and hemodynamic outcomes. It was found that using just static, no-flow conditioning of cultured PSIS valve bioscaffolds with seeded VICs/VECs in a pro-calcific medium was just as effective at creating early-onset CAVD tissue as using high oscillatory flow (*OSI* = 0.5) conditioning. Based on the values obtained for *RF*, *ΔP*, *Q_RMS_*, and *EOA*, via our hydrodynamic functionality testing (Figure 3), none of the valves would be clinically classified as having even mild CAVD [2]. Only Δ*P* showed a peculiar finding of a significantly greater value in the static compared to the raw PSIS and bioreactor compared to static valve groups (Figure 3B). This can be attributed to either issues in the suturing of the bioreactor valve or application of an oscillatory flow over the valve, resulting in reduced overall calcification deposition and thus lower Δ*P*. Nevertheless, the Δ*P* values reported were all <25%—well below the accepted threshold for even mild CAVD [2]. There were no significant differences (*p* > 0.05) between any of the groups for the *RF*, *Q_RMS_*, or *EOA* (Figure 3A,C,D). *EOA* was higher and Δ*P* was lower in the high oscillatory flow culture, bioreactor group (*OSI* = 0.5), compared to the static, no-flow group. We suspect that the high oscillatory flow (*OSI* = 0.5) environment was able to delay the progression of calcific valve disease and suspect this may be the reason early-stage heart valve calcification is seldom detected using solely hydrodynamic functionality measurements.

While these macroscale fluid-based metrics are unable to detect early-stage valve calcification, on the other hand, our nanoindentation results (Figure 4) did show a measurable difference. The Young’s modulus of the cultured valves from both the statically cultured PSIS bioscaffold with seeded VICs/VECs, in pro-calcific media, as well as the bioreactor high-oscillatory-flow-conditioned (*OSI* = 0.5) and cultured PSIS bioscaffold with seeded VICs/VECs, in pro-calcific media, were both very significantly higher (*p* < 0.01), compared to the Young’s modulus of the raw PSIS control valve group. The raw group had a lower Young’s modulus, indicating greater flexibility and ease of bending. In contrast, both calcified groups had slightly higher values, making them stiffer and more resistant to deformation. While all valves maintained high elasticity without impairing hydrodynamics, the calcifications led to morphological changes in the static and bioreactor groups, resulting in less compliant and overall stiffer leaflets. This suggests that the Young’s modulus of a leaflet is a clear, solid, mechanical property that does, at least, present at the early stage of CAVD.

Concurrently, when looking at the hemodynamic results from our FSI simulations, our hydrodynamic calculations matched well with the in vitro results, except for the metrics of *RF* (Table 2). We suspect that this is not due to an issue in the material properties, boundary conditions, or other model settings but rather a limitation of our solver. Surface-to-surface contact of the leaflets can potentially result in splitting of the fluid domain during coaptation, which results in the ALE formulation struggling to converge. Therefore, an artificial gap of ~0.2 mm was enforced between all surfaces, which, while preventing the fluid domain from collapsing, artificially introduced some regurgitant flow and thus greater *RF* for all groups during diastole. Nevertheless, the hemodynamics did show marked differences across valves despite this fault. *WSS* was shown to be lower in the belly region of all valve groups with greater amounts at the free edge and towards the sinus wall (Figure 5). *TAWSS* was statistically significant between the raw and either of the calcified groups; however, there was no significant difference between the calcified groups (Figure 6A). The stiffer valve resulted in greater perfusion of the leaflets as they did not close as smoothly during diastole, which corresponded to higher *WSS* over a cardiac cycle. Although *OSI* remained elevated in all groups (>0.40), it had a slight decrease in the static and bioreactor groups compared to the raw group. However, this difference was not found to be statistically significant (Figure 6B), although further studies with a larger sample size may be required in order to fully confirm this result.

Our study had various limitations which must be noted. Sample size was a major limiting factor in our work with each in vitro experimental group only consisting of three samples with three recordings each for either the hydrodynamic or mechanical tests. Nanoindentation tests were only performed at the center of the sample rather than at multiple points. The heterogeneity of the calcifications would result in varying degrees of stiffness if multiple parts of the samples had been sampled. Only very-early-stage non-symptomatic CAVD was mimicked on the valves due to the time expense of increasing the number of calcified nodules to levels usually associated with symptomatic mild CAVD [37]. Another limitation is the pro-calcific media recipe used; a different method of calcification could alter the results of the experiment by inducing a different propagation of calcific nodule formation [16]. In addition, constant scaffold thickness was assumed for shell modeling per group rather than varying it from base to tip. Our use of artificial numerical dampers to ensure model stability eliminated any high-frequency valve modes, which may have played a role in the more uniform valve coaptation, and thus lower RF, seen in the in vitro results compared to in silico.

## 5. Conclusions

Clinically, there is no difference in hydrodynamics between a healthy aortic valve and one that is in the very early stages of CAVD. We investigated if such a difference could be found using hemodynamics from simulations instead. In our in vitro valve studies between healthy (raw PSIS) and calcified (static/bioreactor-conditioned) groups, we confirmed that our hydrodynamics would be classified as healthy based solely on metrics of *RF*, Δ*P*, *Q_RMS_*, and *EOA*. The Young’s modulus, however, was significantly greater between the raw and either of the calcified groups. This indicates that, while there was not a clinical difference in hydrodynamics, the mechanical properties of the bioscaffolds did undergo a measurable change that affected their flexibility even at the very mild stage of CAVD. Our FSI model analysis also concurrently found that *TAWSS* was increased to a statistically significant level when comparing normal to calcified groups alongside a smaller, non-statistical, drop in *OSI*. Our results suggest that hemodynamics may be used to identify the possible presence of early-stage CAVD when hydrodynamics alone cannot discern a difference.

## Figures and Tables

**Figure 1 bioengineering-11-00955-f001:**
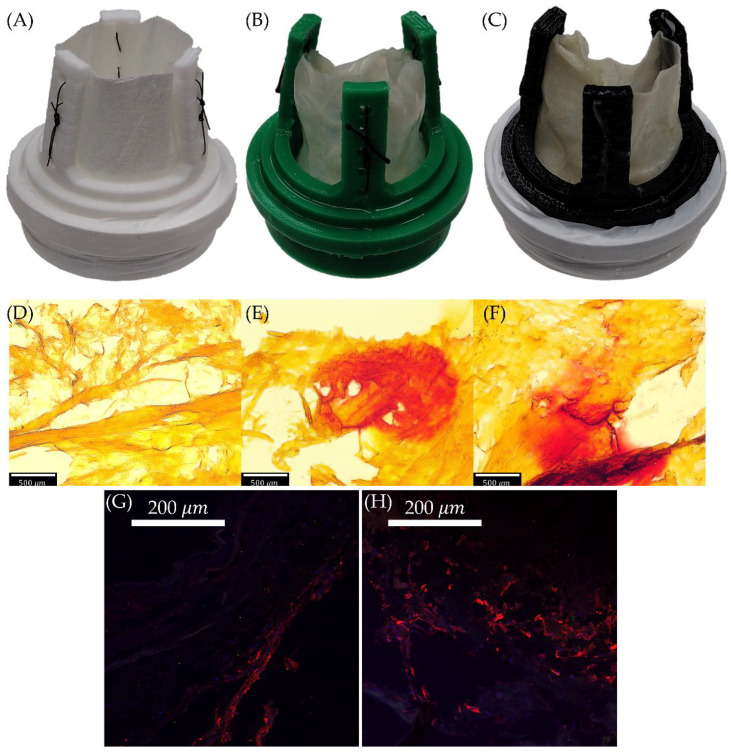
Valve Experimental Groups and Calcium Staining. (**A**–**C**) Valves sutured to 3D printed valve holders Raw (**A**), Static (**B**), and Bioreactor (**C**). (**D**–**F**) Alizarin red staining for the same groups Raw (**D**), Static (**E**), and Bioreactor (**F**). (**G**,**H**) IF stains showing CD31 in green and αSMA in red, Static (**G**), and Bioreactor (**H**). Scale bars are included as dimensional references.

**Figure 2 bioengineering-11-00955-f002:**
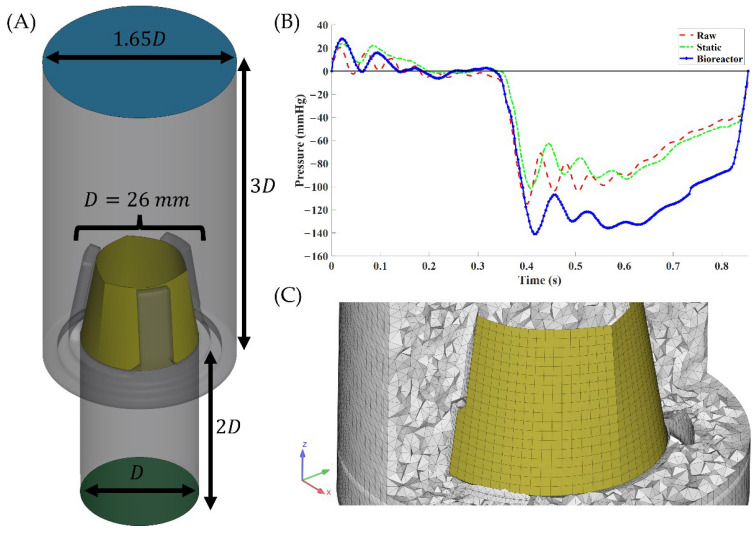
Computational model information. (**A**) Aortic valve conduit and valve holder (gray), scaffold (yellow), inlet (green), and outlet (blue), alongside dimensions scaled to valve diameter, *D*. (**B**) Inlet pressure differential boundary conditions, defined as the difference in pressure between the aortic and ventricle sides of the valve, with raw (dashed red), static (dashed-dotted green), and bioreactor (solid-dotted blue). (**C**) Example of cross-sectional view of finalized volume mesh indicating the mesh resolution and boundary layers inflated along the scaffold.

**Figure 3 bioengineering-11-00955-f003:**
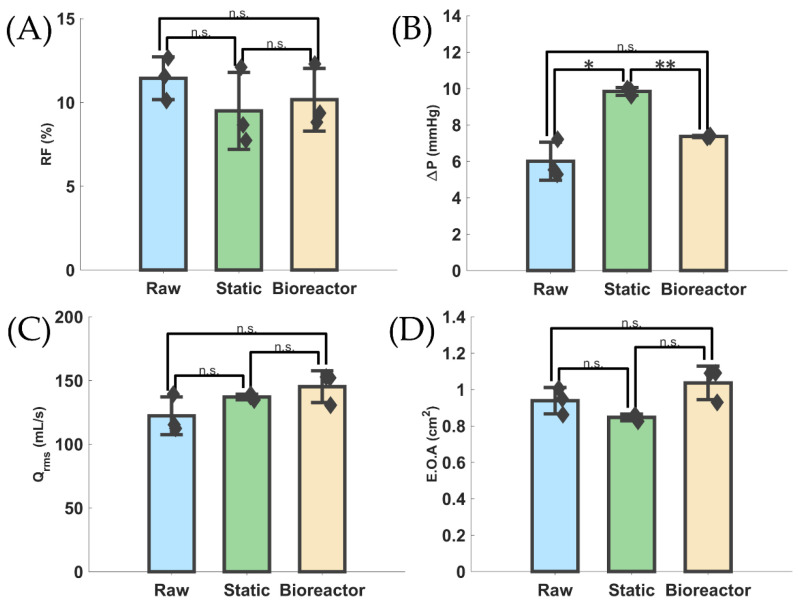
Hydrodynamic results from pulse duplicator. (**A**–**D**) Bar graphs of *RF*, Δ*P*, *Q_RMS_*, and *EOA* between each group with statistical differences highlighted. * indicates a *p*-value < 0.05, ** indicates a *p*-value < 0.01, and n.s. indicates no significant difference, i.e., a *p*-value > 0.05. Diamonds indicate data points.

**Figure 4 bioengineering-11-00955-f004:**
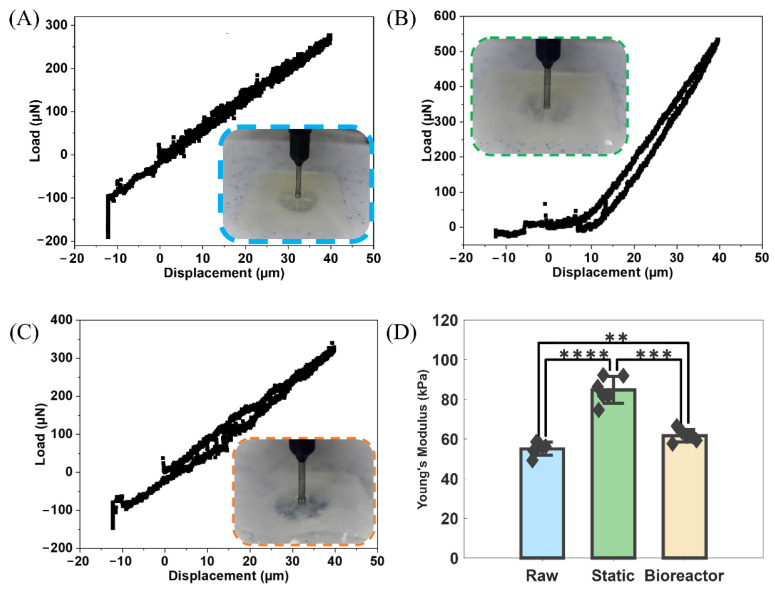
Mechanical results from nanoindentation. (**A**–**C**) Load (μN) vs. displacement (μm) curves for raw, static, and bioreactor PSIS samples, respectively. Overlay shows the nanoindenter head contacting the tissue sample. (**D**) Bar graph of Young’s modulus of the loading curve between each group with statistical differences between groups highlighted. ** indicates a *p*-value < 0.01, *** indicates a *p*-value < 0.001, and **** indicates a *p*-value < 0.0001. Diamonds indicate data points.

**Figure 5 bioengineering-11-00955-f005:**
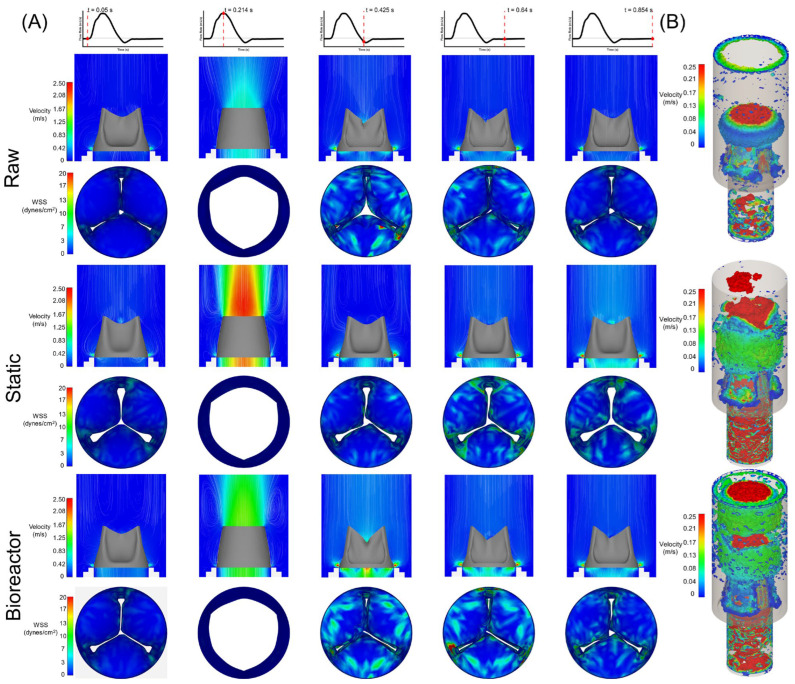
Flow results. (**A**) Velocity and wall shear stress (*WSS*) plots across cardiac cycle for all groups: raw, static, and bioreactor. (**B**) Instantaneous iso-surface of the Q-criterion during peak flow (t≅0.214 s). Simulations were run for three cycles and the results of the third cycle were used in analysis. Results are based on a simulation performed using averaged scaffold thickness and Young’s modulus per group.

**Figure 6 bioengineering-11-00955-f006:**
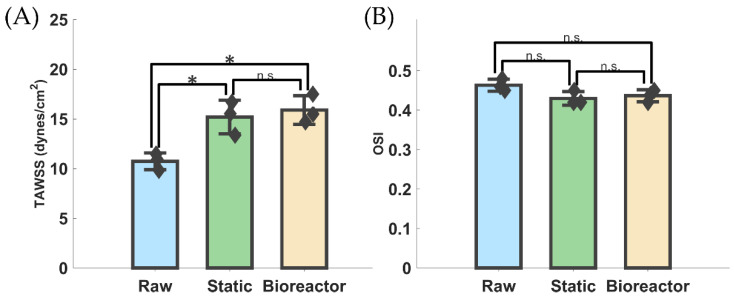
Hemodynamic results from FSI simulations. (**A**,**B**) *TAWSS* and *OSI*, respectively, between each group with statistical differences highlighted. * indicates a *p*-value < 0.05 and n.s. indicates no significant difference, i.e., a *p*-value > 0.05. Diamonds indicate data points.

**Figure 7 bioengineering-11-00955-f007:**
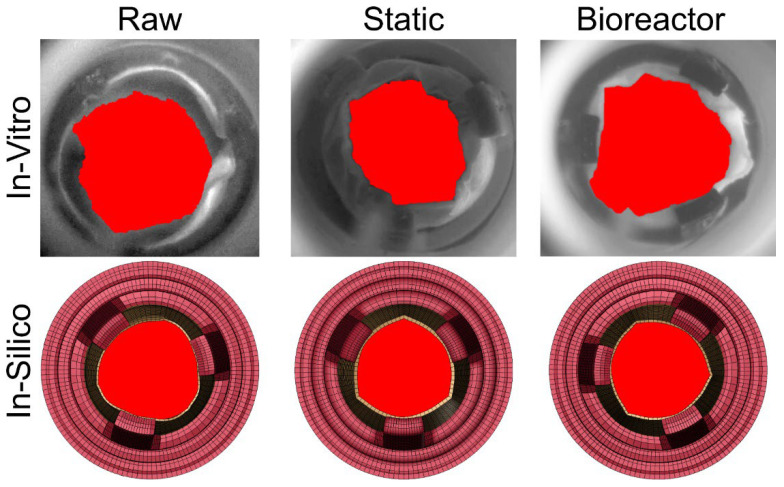
Images used for GOA measurement at concurrent timepoints between in vitro (**top row**) and in silico (**bottom row**) across raw, static, and bioreactor groups. Cross-sectional area is shown in red. For dimensional reference, all valve holders had a base diameter of 26 mm.

**Table 1 bioengineering-11-00955-t001:** Mesh independence study.

Maximum Cell Length (mm)	Number of Elements (Fluid)	Number of Elements (Solid)	Max Velocity		Max Von Mises Stress	
			|*V_Max_*| (m/s)	Percent Diff (%)	Stress (kPa)	Percent Diff (%)
2.00	311,982	648	1.340	12.03	55.95	4.59
1.00	693,294	1440	2.194	0.06	47.49	0.50
0.50	1,386,588	2736	2.185	0.05	47.12	0.30
0.25	2,773,176	5198	2.189	-	46.55	-

Percent difference, $\left|\%∆\right|$, defined as: $\frac{\left|V_{1}-V_{2}\right|}{(V_{1}+V_{2})/2}$.

**Table 2 bioengineering-11-00955-t002:** Hydrodynamic comparisons.

		Raw			Static			Bioreactor		
		In Vitro	In Silico	$\left|\%∆\right|$	In Vitro	In Silico	$\left|\%∆\right|$	In Vitro	In Silico	$\left|\%∆\right|$
RF (%)		10.84	17.14	45.03	9.50	20.56	73.59	10.16	24.56	82.95
ΔP (mmHg)		6.38	8.10	23.76	9.84	11.50	15.56	7.37	8.40	13.06
QRMS (mLs)		127.35	120.81	5.27	137.14	125.13	9.16	145.24	130.89	10.40
$E.O.A\;({cm}^{2})$		0.98	0.82	17.78	0.84	0.72	15.38	1.04	0.88	16.67

Percent difference, $\left|\%∆\right|$, defined as: $\frac{\left|V_{1}-V_{2}\right|}{(V_{1}+V_{2})/2}$.

**Table 3 bioengineering-11-00955-t003:** GOA comparisons.

		In Vitro	In Silico	$\left|\%∆\right|$
		(cm^2^)	(cm^2^)	
Raw		3.24	3.08	5.06
Static		2.26	2.76	19.92
Bioreactor		2.83	2.96	4.49

Percent difference, $\left|\%∆\right|$, defined as: $\frac{\left|V_{1}-V_{2}\right|}{(V_{1}+V_{2})/2}$.

## Data Availability

The data presented in this study are available on request from the corresponding author.

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
