# Peer review of "Computational Model for Early-Stage Aortic Valve Calcification Shows Hemodynamic Biomarkers"

_bioengineering, 2024, doi:10.3390/bioengineering11100955_

Round 1

Reviewer 1 Report

Comments and Suggestions for Authors

This case report is well-structured overall and contributes to current knowledge. However, it requires several modifications, which we will point out in the comments sections below.

Abstract

1.     Some sentences are complex and could be simplified for better comprehension. (line 17-18: "The ability to predict this disease’s likelihood is crucial for proactive patient monitoring before necessitation of aortic valve replacement surgery.")

2.     It should be noted that an unclear group description at the beginning of the article will confuse readers. (line 21)

Materials and Methods

3.     Vascular endothelial cells in vitro were adherent. Could you explain where the cells are in the bioreactor after they attach? How do you ensure no contamination when culturing in an in vitro bioreactor for 7 days? (line 107)

Results, Discussion, and Conclusions

4.     From Figure 3, it seems that only two of the samples in the raw group are inconsistent with the three mentioned above. Please explain this discrepancy.

5.      A lower Young's Modulus indicates a more flexible material that deforms more easily. It is recommended to explain the results of this part in more detail. (line 306 and 383)

6.     Please simplify the complex sentence: "Nonetheless, the Young's Modulus will still significantly greater (p < 0.01) between the raw control and either of the calcified groups, static or bioreactor, indicating while there wasn’t a measurable difference in hydrodynamics the mechanical properties of the bioscaffolds did undergo a change."

Shaping of the Manuscript

7.     The English must be revised. Many sentences need modifications to be comprehensible. Grammatical and typographical errors should be carefully checked (line 365: “this let,” etc.). Please review them thoroughly.

Comments on the Quality of English Language

The English must be revised. Many sentences need modifications to be comprehensible. Grammatical and typographical errors should be carefully checked. Please review them thoroughly.

Reviewer 2 Report

Comments and Suggestions for Authors

Calcific aortic valve disease (CAVD) is a common problem associated with the valve transplantation surgery. To determine the calcification of the aortic valve the haemodynamic evaluation may be the precise approach to determine the early stage CAVD which will further help to treat the problem before showing adverse effects. The in vitro in silico correlation will be beneficial for the further development of machine learning approach to predict the CAVD by analysing hemodynamic biomarkers.

In Hydrodynamic approach of regurgitation values in in silico and in vitro results are dissimilar.

Line 46; The full forms of SAVR and TAVR should be added.

Line 96; The rationale behind the specific ratio of VIC/VEC coculture should be added.

Comments on the Quality of English Language

Abbreviation should be mentioned properly.

Reviewer 3 Report

Comments and Suggestions for Authors

This paper used a PSIS scaffold as a testing platform to identify biomechanics biomarkers of valve calcification. This testing consisted of 3 groups a “raw” group, “static” group which contained VIC and VEC, but no flow, and “bioreactor” group with VIC, VEC, and oscillatory flow. This paper identifies and proposes new biomechanics markers that can potentially be used to identify calcified valves, potentially in its early stages. The implication here being that this methodology could potentially be used to diagnose calcification aortic valve disease before it becomes symptomatic. However, there are quite a few major points that need to be addressed as follows:

While this paper states hemodynamic markers can be used to potentially identify valve calcification it’s not entirely clear what they are. The data shows almost everything reported was statistically insignificant except the pressure difference. Also it’s stated that TAWSS  increases, but the only evidence of this are the color plots in figure 5. Was WSS quantified as was the other hemodynamics values? Meaning was TAWSS differences statistically significant or insignificant? It’s not clear from what’s reported here. All that’s presented are a few color plots of WSS that show higher WSS.

VIC and VEC were grown in pro calcification media for 7 days, but it’s not clear what the cellular density and structure looked like on the PSIS scaffolds at any point during this time. A picture showing the cells actually on the scaffolding would be helpful. Perhaps a day 1 and day 7 or something showing actual cells growing would be appreciated.

As a follow up did the authors do a viability assay? How are the authors confident that most of the cells even survived for so long on the scaffolds?

Valves under all conditions were tested in a 20% saline solution, which is a newtonian, but blood is non-newtonian. The properties of the fluid could have easily been adjusted in vitro. Why not do this ? And how can the authors be sure there results are accurate?

In regards to the newtonian question, the authors could have easily run the same computational simulation using a Non-Newontian fluid model as well. Seems like comparing this to the Newtonian numbers could have given more confidence in the numbers. Again why not run a model with the fluid properties more similar to what the valves will experience in vivo?

Please provide more detail on the differences computationally between the “raw” vs. “static” vs. “bioreactor” models as this is not at all clear. For example it’s clear that that in vitro static and bioreactor models both have cells and the raw does not, but it’s not clear how the authors accounted for these differences in the FSI model. Main point being is that the authors need to explain much better what the difference are between each model….was it geometry and mechanical properties, for example?

As a followup to previous question were the same simulation and mesh parameters used for each model? This is not clear.

Although calcification is reported to appear to be similar in static and bioreactor models, it did exist. With that being said, how did the authors incorporate valve calcification in their models, if at all?

Figure 5 - Multiple issues with this figure. Please note the following bullet points:

 - the text for these plots are extremely small and illegible (even after zooming in at maximum magnification). 

- its not clear at all what the plots above the velocity color plots are in A 

- its not clear what each columns of data represent. Since there are various degrees of valve openings in each set I’m assuming these are different time points? If so please put the time or whatever  the property is that distinguishes each column data set.

Plot or table showing data with GOA measurements would have been helpful

Line 211 “ Young’s modulus from from nano indentation 55.10, 4.79, and 61.72 kPa for raw, static, and bioreactor groups, respectively”. I’m assuming that the “4.79” for static is an error as this number does not match your plots for E.

This study essentially showed that calcified valves are stiffer and WSS is higher in valves with calcification. This is not entirely a surprise here as calcification around valves will increase turbulence leading to a WSS gradient, which would presumably lead to localized increases of WSS around the valve. How or why is this surprising? The authors should provide more detail on the novelty of the methodology and results so that readers can appreciate  the impact of this work.

Can the authors explain how these findings compare to other previously reported data?

Comments on the Quality of English Language

No comments.
